# Long-Term Trajectory and Risk Factors of Healthcare Workers’ Mental Health during COVID-19 Pandemic: A 24 Month Longitudinal Cohort Study

**DOI:** 10.3390/ijerph20054586

**Published:** 2023-03-04

**Authors:** Alice Fattori, Anna Comotti, Sara Mazzaracca, Dario Consonni, Lorenzo Bordini, Elisa Colombo, Paolo Brambilla, Matteo Bonzini

**Affiliations:** 1Department of Clinical Sciences and Community Health, University of Milan, 20122 Milan, Italy; 2Occupational Medicine Unit, Fondazione IRCCS Ca’ Granda Ospedale Maggiore Policlinico, 20122 Milan, Italy; 3Department of Neurosciences and Mental Health, Fondazione IRCCS Ca’ Granda Ospedale Maggiore Policlinico, 20122 Milan, Italy; 4Department of Pathophysiology and Trasplantation, University of Milan, 20122 Milan, Italy

**Keywords:** nurses, physicians, SARS-CoV-2, anxiety, depression, post-traumatic stress, moral distress

## Abstract

Background: Research has shown the substantial impact of the COVID-19 pandemic on healthcare workers’ (HCWs) mental health, however, it mostly relies on data collected during the early stages of COVID-19. The aim of this study is to assess the long-term trajectory of HCWs’ mental health and the associated risk factors. Methods: a longitudinal cohort study was carried out in an Italian hospital. At Time 1 (July 2020–July 2021), 990 HCWs took part in the study and completed the General Health Questionnaire (GHQ-12), the Impact of Event Scale (IES-R), and the General Anxiety Disorder (GAD-7)questionnaire. McNemar’s test measured changes in symptoms’ trajectories, and random effects models evaluated risk factors associated with scores above the cut-off. Results: 310 HCWs participated to the follow-up evaluation (Time 2; July 2021–July 2022). At Time 2, scores above cut-offs were significantly lower (*p* < 0.001) than at Time 1 for all scales (23% vs. 48% for GHQ-12; 11% vs. 25% for IES-R; 15% vs. 23% for GAD-7). Risk factors for psychological impairment were being a nurse (IES-R: OR 4.72, 95% CI 1.71–13.0; GAD-7: OR 2.82, 95% CI 1.44–7.17), a health assistant (IES-R: OR 6.76, 95% CI 1.30–35.1), or having had an infected family member (GHQ-12: OR 1.95, 95% CI 1.01–3.83). Compared to Time 1, gender and experience in COVID-19 units lost significance with psychological symptoms. Conclusions: data over more than 24 months from the pandemic onset showed improvement of HCWs’ mental health; our findings suggested the need to tailor and prioritize preventive actions towards healthcare workforce.

## 1. Introduction

A large amount of literature is now available to attest to the burden of the COVID-19 pandemic on Health Care Workers’ (HCWs) mental health. Early evidence following the pandemic’s onset showed a severe psychological impact resulting in a high level of sleep disorders, anxiety, depressive and post-traumatic symptoms among HCWs [1,2,3,4]. The literature suggested a higher risk for nurses, female, younger and frontline HCWs, while among occupational factors, the time spent working in a COVID-19 area, poor social support, and lack of personal protective equipment (PPE) resulted in an increased risk of adverse mental health outcomes [5,6,7,8]. Research also suggested how personal concerns, such as the fear of being infected or infecting family members, may have caused serious distress, as well as actually being infected or having a family member or friend infected [9,10,11]. Furthermore, different studies specifically dedicated attention to the experience of moral distress and its detrimental effect on psychological health [12].

In addition to the pandemic-related risk factors, HCWs may have been exposed to traditional psychosocial risk factors such as high demands, lack of resources, inadequate staffing, and irregular and prolonged shift works [13]. Since prevalence of occupational stress and burnout in pre pandemic times was substantial [14,15,16,17], it is difficult to establish the effect of the pandemic itself on HCWs’ mental health. 

Most research on the impact of COVID-19 on HCWs’ mental health relies on cross-sectional data collected during the early stages of COVID-19, and longitudinal studies are mainly focused on the first year of the pandemic [18,19,20,21]. Maunder and colleagues (2022) studied the longitudinal trend of burnout and psychological distress among a single cohort of HCWs from the fall of 2020 to the summer of 2021: they found symptoms changing differently over time depending on the professional role, with nurses mostly reporting the highest rates of burnout. Although both distress and burnout decreased after one year, during the summer of 2021, disturbance levels remained higher than pre-pandemic benchmarks. A second prospective cohort study of 22,501 English HCWs [22] conducted from April 2020 to August 2021 showed mental health symptoms varying over the 17 month follow-up, with a higher prevalence during the periods of higher pressure on the healthcare systems. Similar to previous studies, younger workers and female staff as well as workers who had inadequate access to PPE, poor support from colleagues/managers and experience of moral distress were at a higher risk of experiencing mental health disturbances. Contrary to Maunder and colleagues, the English study did not find differences across occupational roles, although ICU workers were more likely to meet the cut-off for post-traumatic stress disorder compared to colleagues of other clinical settings.

### The Italian Context

Italy (particularly Northern Italy) was the first European country to be affected by COVID-19. On 12 March 2020, shortly after the first confirmed SARS-CoV-2 infections, a national strategy for the COVID-19 response was developed and restrictive measures (e.g., national lockdown) were implemented; each region and related public health services developed specific plans according to the allocated resources, facing severe difficulties in fighting an unprecedented healthcare emergency [23,24]. HCWs found themselves to be at high risk of infection and of infecting a family member, to be inadequately informed and unprepared to face the COVID-19 pandemic and experiencing mental health issues [25,26].

Since the onset of the pandemic in February 2020, Italy experienced four major waves of infection (March–July 2020; September–December 2020; February–May 2021; November 2021–March 2022, respectively) with an overall 25,791,569 cases of confirmed SARS-CoV-2 infection and 186,643 COVID-related deaths [27]. 

The COVID-19 vaccine was available from late December 2020. HCWs were among the first to receive the vaccine and Italy was the first European country making vaccination mandatory for healthcare staff. About 90% of the Italian healthcare workforce is now fully vaccinated [28]. 

In the Italian context, available studies carried on until mid-2021 showed different trends in mental health outcomes, with higher rates of anxiety, depression and burnout in 2021 compared to 2020 found by Lasalvia et al. [29] in repeated cross-sectional studies; and a decrease in depression, anxiety, and post-traumatic symptoms, except for increasing insomnia disturbances, revealed by Rossi et al. [30] in a longitudinal cohort study (2021). 

Given the paucity of research on the long-term impact, we are not aware of the quality of HCWs’ mental health during the second year of the pandemic. Although previous pandemics such as SARS may not be comparable to COVID-19 in its clinical, social and psychological widespread impact, results from previous experience showed HCWs might be suffering from long-term psychological disturbances [31]. 

The aim of this study is to assess the long-term trajectory and associated risk factors of anxiety, general discomfort, and post-traumatic symptoms among HCWs through a longitudinal cohort study carried out between July 2020 and July 2022 at a university hospital in northern Italy. 

## 2. Materials and Methods

### 2.1. Study Design and Population

A multi-step process to evaluate and monitor workers’ mental health was developed in June 2020 at a large university hospital in Milan, Italy. The entire process was conducted jointly by the Occupational Medicine and Psychiatry units.

The process aimed to evaluate HCWs’ mental health through a two-level screening and to offer psychological support and psychiatric treatment to those who showed psychological distress. For an extensive description of the methodology adopted for this process, see Fattori et al. [10].

At baseline (from July 2020 to June 2021; Time 1), 990 HCWs took part in the study and completed a multi-step psychological assessment within the occupational health medical surveillance. Results related to the first-year assessment (i.e., Time 1) were published elsewhere [5].

Follow-up data was collected between July 2021 and July 2022 (Time 2). Workers who participated at Time 1 (T1) were invited at Time 2 (T2) through phone calls by Occupational Medicine Unit staff; a subsequent email was sent to reach out to those who did not answer two phone calls. Baseline participants had been informed of the 12-month recall during T1 data collection; as for baseline, follow-up participation was on a voluntary basis.

Occupational physicians within the medical surveillance required by Italian Legislation in terms of occupational safety (i.e., Legislative Decree n.81/2008) performed follow-up evaluation. 

### 2.2. Assessment Measures

During follow-up evaluation, participants completed, for the second time, the Time 1 psychometric scales: –The General Health Questionnaire 12 (GHQ-12; [32]) in the validated Italian version to assess psychological discomfort and short-term changes in mental health [33,34]. For this study, we adopted the dichotomous scoring method (0-0-1-1) and a score equal to or above 4 as the cut-off point [35].–Impact of Event Scale—Revised (IES-R; [36]) to evaluate post-traumatic symptoms specifically related to the COVID-19 emergency; 22 questions explored intrusion, avoidance, and hyperarousal symptoms on a 5 points Likert scale ranging from 0 (not at all) to 4 (extremely). A total score of 33 on the IES-R yielded a diagnostic sensitivity of 0.91 and specificity of 0.82 [37]. We adopted the Italian version, which showed optimal psychometric properties [38].–The Generalized Anxiety Disorder questionnaire (GAD-7; [39]) to screen anxiety symptoms, with a score of 10 or greater representing a cut-off point to identify cases of general anxiety disorder.

Participants were also asked about exposure, concerns and health beliefs related to COVID-19 in the previous six months through multiple choice questions (not at all, little, enough, very). Specifically: having been positive for COVID-19, having family members positive for COVID-19, having fear for their personal safety or fear of infecting family members, experiencing social discrimination for being a HCW or changes in their families’ habits, and having thoughts about changing their job.

Occupational physicians updated and recorded subjects’ occupational data (i.e., working experiences in the COVID-19 area), as well as pathological events and ongoing medical treatments.

### 2.3. Statistical Analysis

We analysed differences between Time 1 and Time 2 measurements (i.e., the total scores considered as quantitative outcomes) using Student’s t test for paired data. In addition, such differences were compared with the “Minimum Clinically Important Difference” (MCID), defined as the smallest change in an outcome perceived as clinically meaningful; MCID was calculated as half the standard deviation of the scores measured at Time 1 [40,41].

Possible differences between groups (gender, age, occupational role, enrollment in COVID-19 units) were graphically explored, representing the average scores for each category at Time 1 and Time 2 and testing mean differences. 

For categorical outcomes (binary response: 0 = scores below cut-off; 1 = scores above cut-off), we applied McNemar’s chi-squared test and calculated odds ratios, risk ratio and risk difference (with central confidence intervals) to analyse paired contingency tables obtained from crossover binary response data. For McNemar’s test only discordant pairs (i.e., whether one measurement is above cut-off and the other one is not) are informative and their ratio gives McNemar’s odd ratio, an estimate of the proportion of change of the outcome variable from one level to the other.

We used random effects logistic models to evaluate variables associated with the probability of scoring above the cut-off. We ran three random effects logistic regressions considering the binary variable indicating whether the score is greater than cut-off or not as outcome. For time-dependent explanatory variables (engagement in COVID-19 area, infection in the previous six months and infected family member in the previous six months), the interaction terms with time were included in the models and if not significant they were removed.

Personal concerns and health beliefs about COVID-19 were graphically explored, distinguishing their distribution at the two different time points. To study their effect on scores in terms of risk factors, they were converted into dichotomous variables (yes = not at all and little; no = enough and very) and put one by one in the random effect models.

## 3. Results

Among the 990 subjects enrolled at baseline (Time 1), 310 (31%) completed the follow-up evaluation (Time 2). The follow-up sample was similar to the baseline sample concerning gender, age and occupational role (Table 1). Participants were predominantly female (71%) and over 40 years old; nurses (i.e., head nurses and licensed nurses; 37%) represented the largest job category followed by physicians (23.5%). Workers engaged in COVID-19 units at Time 2 were a small proportion (13%), while 39% declared a previous experience. Twenty-three subjects (7%) were positive for COVID-19 during the previous six months and 27% had a family member positive for COVID-19 during the previous six months.

Figure 1 shows the percentage of subjects with scores above cut-offs for each questionnaire, calculated according to the total monthly number of participants, whether they were enrolled at Time 1 or 2. Trajectories show the overall decreasing psychological impairment for all the scales adopted from July 2021.

Table 2 presents average scores obtained in the three questionnaires at the two different time points. At Time 2, means were statistically significantly lower than at Time 1 (*p* < 0.001, *t*-test for paired samples), indicating an overall improvement in mental health at all scales. The difference in the two measurements was substantial for IES-R and it may be considered clinically meaningful according to MCID, while GAD-7 showed the smallest decrease. Stratification by groups (age, gender, occupational role, enrollment in COVID-19 units) gave similar decreasing trajectories of average scores for each category and means resulted far from the cut-offs at Time 2 (Appendix A). The associated group means and standard deviations of scale scores are represented in Appendix A.

Scores below or above cut-offs at Time 1 and Time 2 were matched in Table 2, which reports McNemar’s test *p*-values, odd ratios, risk ratios and risk differences. At Time 2 the percentage of overpassing cut-off was lower than at Time 1 for all scales (23% for GHQ-12, 11% for IES-R and 15% for GAD-7). The McNemar’s Chi-square test provided small *p*-values, indicating statistically significant difference in the distributions at Time 1 and Time 2 for all the scales. Few workers scored above cut-off at Time 2 and did not at Time 1 (9% for GHQ-12; 3% for IES-R and 6% for GAD-7); additionally, 34%, 17% and 61% of subjects obtained scores below cut-off of GHQ-12, IES-R and GAD-7, respectively, after scoring above cut-offs at Time 1. Thus, McNemar’s odd ratios were large (3.89 for GHQ-12; 5.30 for IES-R; 2.53 for GAD-7) and their confidence intervals did not contain 1, indicating that the remittent condition (i.e., scoring above cut-off at Time 1 only) was much more plausible than the incident condition (i.e., scoring above cutoff at Time 2 only). Similarly, risks ratio (2.09 for GHQ-12; 2.17 for IES-R; 1.53 for GAD-7) and risks difference (0.25 for GHQ-12; 0.13 for IES-R; 0.08 for GAD-7) showed a higher risk for scoring above the cut-offs at Time 1 than at Time 2.

Random effects models for repeated measures investigated possible risk factors for scoring above cut-off for all scales. Table 3 reports odds ratios and confidence intervals of the logistic regression models. The risk of scoring above cut-offs was significantly lower at Time 2 then at Time 1 (OR = 0.35 for GHQ-12; OR = 0.32 for IES-R; OR = 0.54 for GAD-7, with confidence intervals not containing 1). Nurses and health assistants reported higher risks of scoring above cut-offs than physicians (OR = 4.72 and 6.76 respectively for IES-R; OR = 2.82 and 2.65 respectively for GAD-7). The presence of an infected family member almost doubled the risk of general distress (OR = 1.95 for GHQ-12). 

Females showed greater risk of surpassing the GHQ-12 and GAD-7 cut-offs, and the working experience in a COVID-19 area caused a higher risk of general distress (GHQ-12) and post-traumatic symptoms (IES-R), although without statistically significant differences.

COVID-19-related fears and overall concerns decreased over time, especially worries of infecting family members (Figure 2). After controlling for age, gender, occupational role, working experience in COVID-19 area, COVID-19 infection, personal concerns and health beliefs, a statistically significant risk factor emerged, with higher ORs observed for reported thoughts about changing one’s job and previous experience of moral distress during Time 1 (Table 3). 

## 4. Discussion

In this longitudinal cohort study, we assessed the long-term trajectory and associated risk factors of anxiety, general discomfort and post-traumatic symptoms in a sample of different healthcare professionals working in a large Italian hospital during the COVID-19 pandemic. To our knowledge, this is the first study to report longitudinal data of HCWs’ mental health beyond 24 months from the pandemic onset, assessing psychological symptoms with repetitive measurements. 

Overall, mental health gradually improved over time although with different trajectories depending on the psychological symptoms considered. At Time 2, according to the scales’ cut-offs, we found reduction rates ranging from 8% for GAD to 25% for GHQ; IES-R showed the smallest proportion of scores above the cut-off compared to GHQ-12 and GAD-7, suggesting a general remission for post-traumatic symptoms related to the COVID-19 pandemic. These results may be consistent with existing literature on long-term psychological disturbances in disaster-affected populations, indicating a gradual remission in disorders albeit with differences, as post-traumatic symptoms generally decrease in the years following the traumatic event while anxiety and depression can persist further [42]. 

Follow-up data collection was conducted after the third wave and during the fourth (Omicron) wave of contagions. During these phases of the COVID-19 pandemic, related hospital admissions and deaths declined considerably, both in the northern regions and throughout Italy [43,44]. Moreover, HCWs had already received (i.e., first trimester 2021) COVID-19 vaccines, as a massive and rapid campaign of health workers’ vaccination occurred between January and February 2021. All these circumstances may suggest a reduction in the workload pressure as well as in the occupational risk for COVID-19 infection. Such changes in workload and in COVID-19-related scenarios, together with expectations of further improvement, may have positively affected HCWs’ mental health [45] and could partially explain the observed reduction in worries and concerns (Figure 2). Although previous research reported an increased intention to leave one’s job by HCWs during the COVID-19 pandemic [46], our results showed a small and constant proportion of subjects who thought about changing jobs (Figure 2).

Interestingly, in our study the most relevant decrease was observed for IES-R, whose items specifically asked participants for post-traumatic responses to the COVID-19 emergency, whereas both GAD-7 and GHQ-12 assessed symptoms, which may be related to broader distressing episodes or to circumstances beyond the pandemic (e.g., “Becoming easily annoyed or irritable”, “Loss of sleep over worry”, “Felt constantly under strain”, etc.).

Although psychological improvement in our sample was substantial, about 10% of participants had persistent impairment and a smaller proportion (from 3% for IES-R to 9% for GHQ-12) experienced discomfort only in the second year, suggesting possible late onset of mental distress. To this regard, it is essential to maintain ongoing mental health monitoring to support long-lasting suffering as well as to prevent late-onset mental illness. In addition to psychological and psychiatric support, HCWs’ wellbeing could be sustained through organizational intervention, such as reducing shift lengths, providing effective training for unfamiliar tasks, safety procedures and stress management [47]. 

Most previous follow-ups conducted in Italian hospitals after 12–15 months from the pandemic onset (i.e., spring 2021) had already shown an overall decrease in distress, anxiety, depressive and post-traumatic symptoms [30,48]; although a study conducted in the same period found an increase in anxiety, depression and burnout [29]. Such differences may be partially explained by considering the geographic area in which these studies were conducted, as the COVID-19 pandemic outbreak was particularly severe for rates of transmission and mortality in northern Italian regions, where Lasalvia and colleagues conducted their study [49].

The lack of literature on HCWs’ mental health during the second year of the COVID-19 pandemic hampers the comparison between our sample and others from different hospitals and countries. To this regard, it is fundamental to consider studies conducted with similar time points (e.g., similar lockdown conditions, viral transmission, and death rates) since mental health impairment may increase during periods with stricter measures and higher death/incidence rates [50]. 

Regarding the risk factors for psychological symptoms, we found different results when considering both Time 1 and Time 2 outcomes compared to Time 1 outcomes only. Being female and younger were not significant risk factors for impaired mental health in contrast with previous results and pandemic-related literature, which showed women at higher risk for mental health disturbances and quality of life disruption beyond occupational role [5,8,51,52]; however, we should also take into account that women were likely to experience an extra burden related to family care during the early stages of the pandemic, and that such overload may have been reduced at Time 2, thus contributing to a decrease in fatigue [53].

Differently from Time 1, working experiences in COVID-19 areas did not increase the risk for psychological impairment. This is consistent with research conducted during the second year of the pandemic, showing that working in COVID-19 areas might no longer be a massive stressor for health impairment, although workers were exposed to adverse conditions such as high emotional demands [54,55]. This may be the result of multifactorial processes such as psychological adjustment, more functional coping strategies, better within-team support, workers’ self-selection, enhanced medical resources to treat COVID-19, etc. In our study, concerns related to COVID-19, albeit decreased during the second year of the pandemic, were still significantly associated with all psychological outcomes. In particular, the experience of moral distress at Time 1 still had significant strong associations with mental health impairment, especially for post-traumatic stress response (OR = 17.7). This is consistent with the literature on moral distress experienced by HCWs during the COVID-19 pandemic, reinforcing the need for urgent interventions to sustain recovery as well as to promote strategies and resources to cope with such stressful situations [12,56,57].

Our study has several limitations. Participation rate at Time 2 was lower than at Time 1 and this may entail a self-selection bias. However, follow-up rates did not differ among subgroups as both baseline and follow-up samples were similar by gender, age and occupational role (Table 1). We should also consider that the reduced proportion of workers who decided to undergo a specific evaluation of their mental well-being might be itself an indirect hint of a general improvement of mental health and of a decrease in COVID-19-related concerns among Italian HCWs.

Another important limitation is the lack of data on the pre-pandemic mental health of our sample; this hampers the estimate of the pandemic impact as the literature suggests that a history of mental disorder may be a prominent risk factor for impaired mental health during pandemics [11,58]. Finally, self-report measures expose our study to potential common-method bias, although we attempted to minimize this risk by grounding data collection during the occupational physician health surveillance (to limit incomplete answers) and by also assessing “objective” data related to COVID-19 exposure (e.g., duration of enrolment in COVID-19 unit, swab results).

Notwithstanding the declared limitations, our findings from data collected in the second year of the pandemic through a longitudinal cohort study may be of great interest. We found that mental health among HCWs gradually improved compared to the previous year; nonetheless, being a nurse or a health assistant, having a family member positive with COVID-19 in the previous six months and previous experience of moral distress still acted as significant risk factors for adverse psychological outcomes.

## 5. Conclusions

Our results showed the need to tailor and prioritize preventive actions towards the healthcare workforce, as nurses and health assistant may be at a greater risk of developing impaired mental health conditions compared to colleagues. This result is coherent with data from previous studies conducted during COVID-19 pandemic, suggesting these professionals experienced persistent fatigue and burden (5, 9). Beyond the pandemic, it is essential to carry on preventive interventions in the healthcare sector through a multi-level approach, as the literature has shown that HCWs experienced chronic exposure to occupational stress and substantial psychological impairment also in the pre-pandemic era [15,17,59]; to this regard, the COVID-19 pandemic worsened pre-existing adverse working conditions, such as understaffing, lack of specialised staff and high workloads [60]. According to our results, possible interventions should consider a wide range of activities aimed at primary, secondary and tertiary prevention to foster both mental health and recovery from illness. Examples may include training programmes to enhance personal resources to cope with stress, to normalize psychological and emotional reaction to public health emergencies, as well as long-term follow-ups to offer ongoing psychological support even after the most critical phases of the pandemic [61,62,63,64]. 

Further results will provide better insights into long-term symptoms’ trajectories and interventions’ effectiveness [65]. Longitudinal studies may help occupational health and safety professional to address all levels of prevention, sustain an ongoing monitoring of mental health, facilitate early identification of impairment and support workers with persistent distress.

## Figures and Tables

**Figure 1 ijerph-20-04586-f001:**
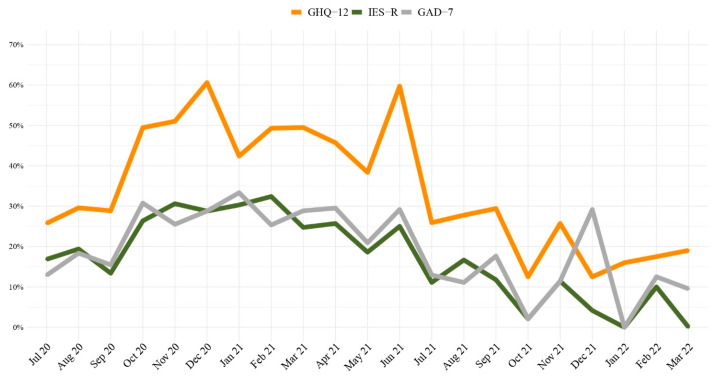
Percentage of subjects overpassing cut-offs by month. Data from April to July 2022 were not represented, as there were few subjects (N < 10).

**Figure 2 ijerph-20-04586-f002:**
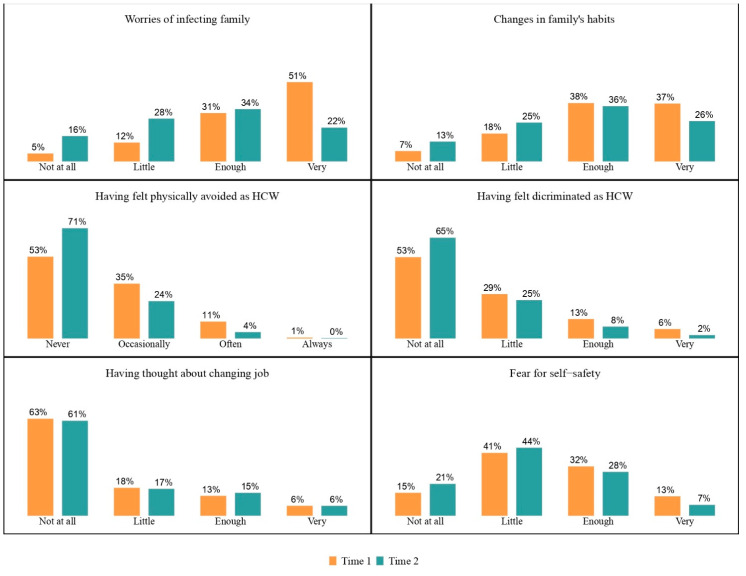
Health beliefs and COVID-19-related concerns: percentage of each answer at Time 1 (orange columns) and Time 2 (green columns).

**Table 1 ijerph-20-04586-t001:** Baseline (N = 990) and follow-up (N = 310) samples characteristics: frequencies and percentages by gender, age, occupational role, COVID-19 area, personal (positive nasopharyngeal swap) and family member infection for COVID-19, worries and health believes related to COVID-19.

	Baseline Sample (N = 990)	Follow-Up Sample (N = 310)	
**Gender**			
Male	297 (30%)	91 (29%)	
Female	693 (70%)	219 (71%)	
**Age group**			
20–29	137 (14%)	36 (12%)	
30–39	276 (28%)	89 (29%)	
40–49	245 (24.5%)	75 (24%)	
≥50	332 (33.5%)	110 (35%)	
**Occupational role**			
Administrative staff	119 (12%)	46 (15%)	
Health assistant	63 (6.5%)	15 (5%)	
Nurse	416 (42%)	116 (37%)	
Physician	233 (23.5%)	83 (27%)	
Others	159 (16%)	50 (16%)	
**Time-dependent covariates**		**Time 1**	**Time 2**
**COVID-19 area working experience**			
No	544 (55%)	151 (49%)	150 (48%)
Previously	202 (20%)	62 (20%)	120 (39%)
Currently	244 (25%)	97 (31%)	40 (13%)
**Positive nasoph. swap**			
Yes	153 (15%)	45 (15%)	23 (7%)
No	837 (85%)	285 (85%)	307 (93%)
**Family member positive to COVID-19**			
Yes	209 (21%)	65 (21%)	84 (27%)
No	781 (79%)	265 (79%)	246 (73%)
Worries of infecting family	792 (80%)	256 (83%)	173 (56%)
Change family habits	695 (70%)	233 (75%)	192 (62%)
Fear for ones’ safety	445 (45%)	138 (45%)	110 (35%)
Having felt discriminated as HCW	179 (18%)	58 (19%)	31 (10%)
Having felt physically avoided as HCW	111 (11%)	36 (11%)	14 (4.5%)
Having thought about changing job	175 (18%)	60 (19%)	66 (21%)
Moral distress	174 (18%)	64 (21%)	

**Table 2 ijerph-20-04586-t002:** Comparison among GHQ-12, IES-R and GAD-7 scores at Time 1 and Time 2. Total scores summary statistics (mean, sd) with paired t-test for mean difference, average absolute difference Δ and sample MCID. Pair-matched table of frequencies and percentages of scores above the cut-offs, with McNemar Chi-square test, McNemar odd ratio (OR), risk ratio (RR) and risk difference (RD) with corresponding 95% confidence intervals.

	Mean (sd) Time 1	Mean (sd) Time 2	|Δ|	Sample MCID	Pair-Matched Table N (%)				McNemar OR (95% CI)	RR (95% CI)	RD (95% CI)
**GHQ-12**	3.7 (3.2)	2.0 (2.7)	1.7	1.6		**Above cut-off** **Time 2**	**Below cut-off** **Time 2**	**TOT**	3.89 (2.53, 6.18)	2.09 (1.66, 2.66)	0.25 (0.19, 0.32)
					**Above cut-off** **Time 1**	44 (14%)	105 (34%)	149 (48%)			
	*Paired t-test: p < 0.001*				**Below cut-off** **Time 1**	27 (9%)	134 (43%)	161 (52%)			
					**TOT**	71 (23%)	239 (77%)	**310**			
					*McNemar Chi-square test: p < 0.001*						
**IES-R**	21.9 (16.9)	13.2 (13.1)	8.7	8.4		**Above cut-off Time 2**	**Below cut-off Time 2**	**TOT**	5.30 (2.67, 11.7)	2.27 (1.59, 3.36)	0.13 (0.09, 0.19)
					**Above cut-off Time 1**	23 (8%)	53 (17%)	76 (25%)			
	*Paired t-test: p < 0.001*				**Below cut-off Time 1**	10 (3%)	224 (72%)	234 (75%)			
					**TOT**	33 (11%)	277 (89%)	**310**			
					*McNemar Chi-square test: p < 0.001*						
**GAD-7**	6.5 (5.2)	5.0 (3.91)	1.5	2.6		**Above cut-off** **Time 2**	**Below cut-off Time 2**	**TOT**	2.53 (1.41, 4.73)	1.53 (1.12, 2.19)	0.08 (0.03, 0.13)
					**Above cut-off Time 1**	29 (9%)	43 (14%)	72 (23%)			
	*Paired t-test: p < 0.001*				**Below cut-off Time 1**	17 (6%)	221 (71%)	238 (77%)			
					**TOT**	46 (15%)	264 (85%)	**310**			
					*McNemar Chi-square test: p < 0.001*						

**Table 3 ijerph-20-04586-t003:** Risk factor analysis through random effects models on the follow-up sample (N = 310): adjusted ORs for scoring above the cut-offs of GHQ-12, IES-R and GAD-7 with associated 95% confidence intervals.

	GHQ-12 OR (95% CI)	IES-R OR (95% CI)	GAD-7 OR (95% CI)
**Gender**			
Male	1	1	1
Female	1.29 (0.83, 2.02)	0.90 (0.41, 1.97)	2.26 (0.86, 2.52)
**Age Group**			
≥50	1	1	1
20–29	1.07 (0.51, 2.25)	0.35 (0.09, 1.42)	0.51 (0.10, 2.52)
30–39	1.28 (0.74, 2.22)	0.79 (0.29, 2.11)	1.01 (0.47, 4.34)
40–49	0.86 (0.49, 1.50)	0.44 (0.16, 1.26)	1.11 (0.36, 3.49)
**Occupational role**			
Physician	1	1	1
Administrative staff	0.56 (0.26, 1.19)	0.51 (0.12, 2.18)	0.50 (0.11, 2.20)
Nurse	1.15 (0.68, 1.94)	4.72 (1.71, 13.0)	2.82 (1.44, 7.17)
Health assistant	0.59 (0.21, 1.66)	6.76 (1.30, 35.1)	2.65 (0.52, 13.5)
Others	1.17 (0.60, 2.29)	2.43 (0.74, 7.92)	2.27 (0.57, 9.04)
**COVID-19 area working experience**			
No	1	1	1
Previously	1.28 (0.64, 2.45)	1.50 (0.49, 4.61)	0.65 (0.23, 1.85)
Currently	1.37 (0.73, 2.45)	1.36 (0.49, 3.77)	0.47 (0.15, 1.45)
**Positive nasoph. swab**	0.74 (0.34, 1.59)	1.07 (0.35, 3.28)	0.44 (0.14, 1.42)
**Family member positive to COVID-19**	1.95 (1.01, 3.83) *	0.88 (0.32, 2.45)	1.18 (0.51, 2.73)
**Time 2**	0.27 (0.18, 0.41)	0.20 (0.07, 0.66)	0.31 (0.16, 0.61)
**Health beliefs and concerns ****			
Worries of infecting family	1.97 (1.24, 3.13)	3.70 (1.59, 8.61)	2.63 (1.05, 6.61)
Change family habits	3.09 (1.90, 5.03)	8.08 (3.21, 20.3)	6.88 (2.62, 18.0)
Fear for ones’ safety	3.83 (2.21, 6.65) *	5.94 (2.59, 13.6)	5.67 (2.85, 11.3)
Having felt discriminated as HCW	2.43 (1.38, 4.29)	7.12 (2.46, 20.5)	3.15 (1.28, 7.71)
Having felt physically avoided as HCW	1.76 (0.87, 3.55)	5.26 (1.66, 16.6)	4.00 (1.17, 13.7)
Having thought about changing job	17.2 (6.57, 45.1) *	19.5 (3.86, 72.4)	19.3 (5.48, 67.6)
Moral distress	3.47 (2.09, 5.76)	17.7 (4.59, 38.2)	5.95 (2.37, 14.9)

* interaction with time is significant. ** put one by one in the model. Reference category: subject answering “No”.

## Data Availability

The dataset generated and analysed during the current study are not publicly available due to restrictions related to our internal review board and to our hospital policy in relations to public health workers. For ethical reasons, to avoid any possible workers’ identification, data are available only in aggregate format upon reasonable request to the Principal Investigator (matteo.bonzini@unimi.it).

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
