# Peer review of "Long-Term Trajectory and Risk Factors of Healthcare Workers’ Mental Health during COVID-19 Pandemic: A 24 Month Longitudinal Cohort Study"

_ijerph, 2023, doi:10.3390/ijerph20054586_

Round 1

Reviewer 1 Report

This is a significant and well-designed study that highlights the importance of longitudinal follow-up in the psychological well-being of HCWs following a pandemic or disaster. Can you comment on your recruitment process? Data on baseline or pre-existing mental illness before T1 would have been helpful since we know this is a significant RF for continued psychological distress. What do you believe led to the significant drop out rate for participants at T2 (follow up sample) and can you comment on which groups were mostly impacted by lower sample size in T2 and why? Did some people leave the profession btw baseline and f/u? Could that have impacted the results? 

What recommendations would you suggest that health care institutions or funders consider when desgning targeted interventions given your results...(e.g timing, content, format), and should family members be included in these interventions given impact on family habits and fears of infecting family members? 

I would strongly recommend that the authors have a native English reader proofread the text thoroughly for grammatical errors and improved language style. There are multiple errors noted throughout.

Author Response

Please find attached the detailed form

Reviewer 2 Report

Manuscript ID ijerph-2187768 entitled " Long-term trajectory and risk factors of healthcare workers’ mental health during Covid-19 pandemic: a 24 month  longitudinal cohort study "

Thank you for the opportunity to review the manuscript with the title “Long-term trajectory and risk factors of healthcare workers’ mental health during Covid-19 pandemic: a 24 month longitudinal cohort study". This study is an interesting and important contribution to the literature that seeks to report on a rigorous study examining the mental health consequences of the Covid-19 pandemic in healthcare providers. Overall, I enjoyed the comprehensive introduction, the sound methodology, the well-structured presentation of the results as well as the discussion of potential implication, strengths and limitations.

Introduction
Cultural considerations. Please refer to the cultural context in a brief subsection

National strategies to respond to the pandemic, nationwide measures, information on the structure and the challenges of the National Health Care System, trust in protocols and hospital management, guidelines released by the national health commission, brief statistics regarding the impact of the pandemic in your country as well as education of HCWs regarding the present and previous pandemics

Methodology

When you refer to nurses, do you include nursing assistants? please provide brief information on the levels of nursing education in your country.

Discussion

A main results of your study is that “risk factors for psychological impairment was being a nurse” Could you please discuss this with reference to literature (e.g. 10.1111/ppc.12946).

Please provide some speculations on across country differences in long term trajectories among HCWs in your discussion section. 

Line 130 an health assistant Please correct

Thank you! All the best!

Author Response

Please find attached the detailed form
